# Feature Disentanglement to Aid Imaging Biomarker Characterization for Genetic Mutations

**Padmaja Jonnalagedda** [1]                                    SJONN002@UCR.EDU
**Brent Weinberg** [2]                                  BRENT.D.WEINBERG@EMORY.EDU
**Jason Allen** [2]                                      JASON.W.ALLEN@EMORY.EDU
**Bir Bhanu** [1]                                        BHANU@VISLAB.UCR.EDU

[1] *Dept. of Electrical and Computer Engineering, University of California Riverside (USA)*

[2] *Dept. of Radiology, Emory University, Atlanta, Georgia (USA)*

## Abstract

Various mutations have been shown to correlate with prognosis of High-Grade Glioma (Glioblastoma). Overall prognostic assessment requires analysis of multiple modalities: imaging, molecular and clinical. To optimize this assessment pipeline, this paper develops the first deep learning-based system that uses MRI data to predict 19/20 co-gain, a mutation that indicates median survival. It addresses two key challenges when dealing with deep learning algorithms and medical data: lack of data and high data imbalance. To tackle these challenges, we propose a unified approach that consists of a Feature Disentanglement based Generative Adversarial Network (FeaD-GAN) for generating synthetic images. FeaD-GAN projects disentangled features into a high dimensional space and re-samples them from a pseudo-large data distribution to generate synthetic images from very limited data. A thorough analysis is performed to (a) characterize aspects of visual manifestation of 19/20 co-gain to demonstrate the effectiveness of FeaD-GAN and (b) demonstrate that not only do the imaging biomarkers of 19/20 co-gain exist, but also that they are reproducible.

**Keywords:** Generative Adversarial Networks, Brain Tumor, Magnetic Resonance Imaging, Limited Dataset

## 1. Introduction

Glioblastoma multiforme (GBM) is the most common and aggressive form of malignant tumor, comprising of 54% of all primary brain tumors (Tamimi, 2017), reporting a 5-year survival rate of 5% (Tamimi, 2017). Assessment of overall clinical outcomes typically requires a combination of clinical, molecular and multi-modal imaging data. This process is time consuming, complex and overloads the clinical work-force. These pitfalls are exacerbated by the increasing incidence of GBM, introduction of high-resolution imaging, scarcity of resources for molecular testing, lack of clinical follow-up and inconsistent data recording across modalities, just to name a few. For prognosis and recurrence estimation, molecular testing is the most precise approach. The concurrent gain of 19/20 chromosomes (19/20 cogain) has been shown to be correlated to prognosis (Geisenberger et al., 2015). Assessment of the mutation using imaging would make the process non-invasive, save resources (human effort, money, time) in treatment planning and also aid with post-treatment understanding (Kanas et al., 2017). Consequently, a hypothesis that is gaining traction is that molecular biomarkers emerge as visual manifestations at a macroscopic level in various imaging

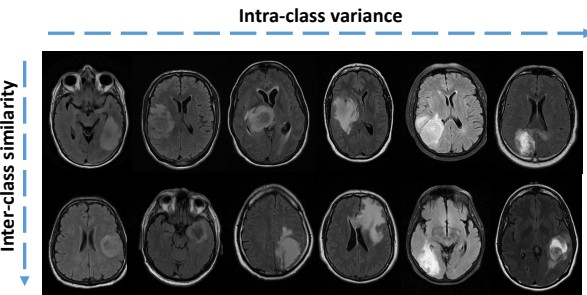

Figure 1: Intra-class variability and inter-class similarity in 19/20 co-gain mutation. Images are all shown in FLAIR modality. Top: No mutation, Bottom: Mutation present

modalities of GBM testing. To this end, the proposed approach attempts to answer the following questions in this paper: (1) Does 19/20 co-gain have a visual manifestation in any of the MR Imaging Modalities? (2) If yes, are these imaging features consistent and reproducible? (3) Which macroscopic features does the mutation present itself in?

During the above assessment, some common yet significant problems were encountered. There is a high inter-class similarity and intra-class variation between control and mutated tumors, as can be seen in Figure 1. Additionally, due to the relatively scarce nature of the mutation, we are faced with a lack of data and a high data imbalance between the two classes with and without mutation. For data imbalance, Generative Adversarial Networks (GANs) have recently been used to learn data distribution to generate diverse samples. Building on that, we could potentially use them in learning the visual indicators of 19/20 co-gain and demonstrate them to be consistent and reproducible. Nevertheless, GANs are deep networks that require a fair amount of data to train and the lack of data in our application renders traditional GANs less than optimal. Thus, we propose a new approach to GANs: FeaD-GAN, a GAN framework based on feature disentanglement to generate synthetic data from very small datasets. FeaD-GAN extricates attributes of the input image into shape and texture and performs latent space recasting using disentanglement and performs re-sampling, thereby increasing the *apparent size* of the dataset. Due to the nature of the dataset, mode collapse is a significant problem. Data driven embedding has been proposed in literature (Xiao et al., 2018) to avoid mode collapse. However, in addition to tackling mode collapse, our goal here is to generate diverse synthetic images from very limited and imbalanced dataset. The independent feature channels provide a larger search space and added control over generated data by influencing how each feature is sampled before embedding, thereby allowing us to tackle all the issues. Achieving stability in GANs is always a challenge, and in this particular case, parallels can be drawn with multi-objective optimization problem  stability and expansive search space. We want to optimize the solution without loss in performance, while training the network end-to-end. The applications of such an approach potentially go beyond brain tumors, for example, anomaly detection problems, data with high class imbalance as well as small datasets  i.e., many real-world datasets.

### 1.1. Contributions

The contributions of our proposed research are as follows:

- A GAN framework effective for learning unapparent discriminatory features while training on very limited data,

- Feature disentanglement and feature recasting for "informative noise" input to GAN for learning subtle discriminatory features,

- Quantitative evidence of presence and reproducibility of visual manifestation of 19/20 co-gain in MR Images across datasets,

- Analysis of three potential macroscopic indicators of mutation: shape, location and texture of the tumor.

## 2. Related Work

Deep learning researchers have analyzed imaging features of some mutations (Kanas et al., 2017; Li et al., 2017; Chang et al., 2018; Lu et al., 2018; Fathi Kazerooni et al., 2019) and have reported positive trends. For learning underlying data distribution and synthesis of diverse samples, GANs have been shown to perform very well (Bowles et al., 2018; Frid-Adar et al., 2018). Since unapparent visual manifestations are being analyzed, it is critical that the tumor features be generated with sufficient detail. For High and Low Grade Gliomas, there has been some research in generating images for data augmentation (Han et al., 2019, 2018). Other than GANs, there are many other data augmentation techniques such as rotation, cropping, etc. The drawback is that these augmentation techniques re-sample from the same overall distribution, without using domain knowledge to recreate significant features that can improve the quality of learning. Some researchers have attempted to use manifold learning (Khayatkhoei et al., 2018), attention-based learning (Zhang et al., 2018), VAEs (Higgins et al., 2017), etc. to improve the quality of learning. In videos, there is evidence that disentanglement helps in better feature learning (Tulyakov et al., 2018). While these are impressive strides in improving quality of learning, most real world data resides in a very high dimensional space and consists of many attributes operating together, which need to be studied and dealt with accordingly. The proposed framework demonstrates the impact of disentanglement for recreating detailed tumor properties that represent mutation information and generate high quality images.

## 3. Proposed Approach

The protocol for demonstrating presence, characterization and reproducibility of features is discussed in this section. Details of the FeaD-GAN framework and how it learns desired data model using limited data are given in section 3.2.

### 3.1. Description of Framework

In the process of detecting imaging biomarkers of 19/20 co-gain and assessing their characteristics, we demonstrate that visual indicators of the mutation exist, that they are consis-

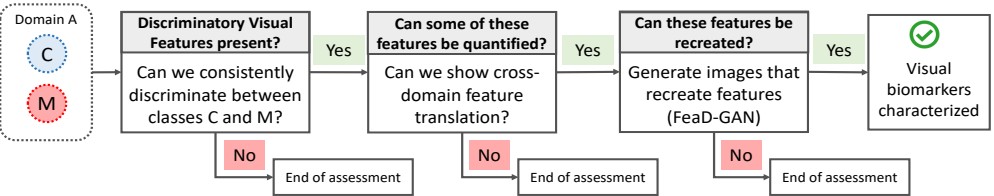

Figure 2: Overall Assessment Pipeline for Biomarker Discovery of 19/20 co-gain. C: Control class and M: Mutated class of images.

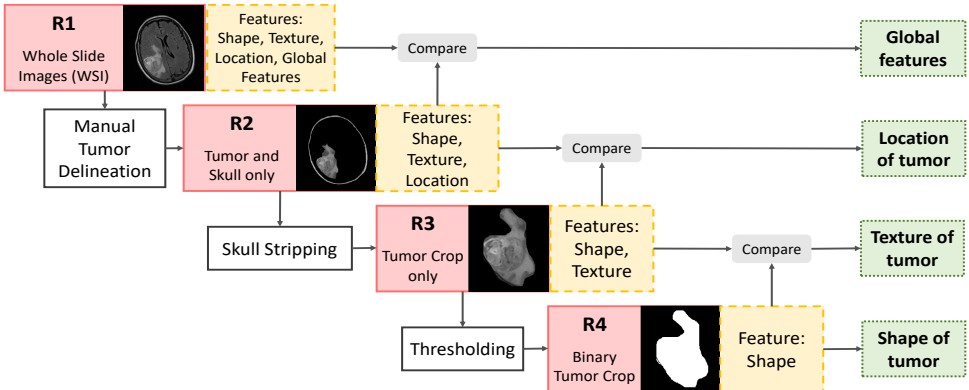

Figure 3: The four data representations (R's) used in biomarker detection. Red/solid boxes: R1-R4 representations; yellow/dashed: features represented in R1-R4; green/dotted: features quantized

tent and reproducible. Each of these processes is described in this section. An overview of our framework is shown in Figure 2.

**Detecting presence of mutation.** The presence of mutation biomarkers is, yet, unapparent to the human eye. We turn to deep learning classifiers for solving this detection problem. Once we can consistently detect the presence of imaging biomarkers using the classifiers, we characterize the biomarkers by quantifying the impact of some macroscopic features.

**Characterization of biomarkers.** The features we will assess in the scope of this paper are: location of tumor, shape of tumor, texture of tumor. To isolate these features, we perform semi-automatic hierarchical feature stripping. The original Whole Slide Images (WSIs) (R1) contain all information about the tumor and its surroundings. The second level of tumor representation (R2 in Figure 3) is delineated tumor with no skull stripping. The tumor boundary is manually delineated. We preferred manual delineation over automated because automated segmentation of brain tumors is a (rapidly) growing field which is yet to

be perfected, and any annotation errors will trickle down to the rest of the model, and thus, affect the analysis of both: classifier and FeaD-GAN. Additionally, there are some benefits of pre-identified ROI (Chaddad et al., 2019). The difference between representations R1 and R2 is that R1 contains global features such as the brain matter, as shown in Figure 3 as opposed to the Shape + Texture + Location in R2. The location is with respect to the skull boundary. R3 is a depiction of the tumor alone. We use a skull stripping algorithm using connected components to convert R2 to R3. Thus, the R3 representation is now stripped of location information. The fourth and final representation R4 is a binary mask of R3, i.e., shape of the tumor. R4 no longer contains texture information. Evaluating the performance of each representation in detecting co-gain, we obtain an estimate of the discriminatory properties held by the corresponding macroscopic feature set. We then compare the performance of these representations with each other to quantify the impact of individual features on co-gain manifestation and subsequently the discriminatory properties they represent. The features analyzed using R1-R4 are shown in Figure 3. At this stage of the analysis, the presence, consistency and characterization of visual mutation indicators are established. The next step assesses whether these indicators are reproducible.

**Reproducibility.** Recreating the visual presence of mutations is of significant value due to its importance in validation of unapparent feature manifestation. Knowing there is a discriminatory feature set to learn from, we can fortify many medical datasets that lack samples. Furthermore, it helps with data augmentation. To reproduce these features, we generate synthetic tumors using FeaD-GAN. To ensure that these synthetic tumors indeed replicate the required feature set, we use the synthetic images only to predict mutation in real inputs. In the next sub-section, we detail the framework of FeaD-GAN.

### 3.2. FeaD-GAN

The overall framework of FeaD-GAN is shown in Figure 4. The motivation behind FeaD-GAN comes from the challenges in this dataset: limited data, high inter-class similarity and intra-class diversity between presence and absence of mutation ( Figure 1) and lack of knowledge of participating feature set. Therefore, to be able to faithfully replicate the subtle differences in features, we propose generating tumors using disentanglement of the imaging features (via FeaD-GAN). Leveraging the observations from Section 3.1, FeaD-GAN performs feature disentanglement to split the tumors into their shape and textures. The shape and texture are both, then, projected onto a latent space. Noise is added to this latent space and from the subsequent probability distribution, we sample data points to recreate the tumor. Thus, FeaD-GAN's input is essentially an educated noise as opposed to random noise in traditional GANs.

**Details of Algorithm.** First step in FeaD-GAN is to accurately obtain shape and texture representations. A binary threshold on the input tumor crop (X) outputs a shape image ($X_\alpha$). This vector is projected onto the latent space using a linear layer followed by ReLU activation. We project this vector to a length of 128 in the latent space. To obtain the texture vector ($X_\beta$), a Gabor CNN (GCNN) is used followed by a similar linear layer and activation. We decided to use GCNN instead of standard CNNs to ensure texture representation of data. Traditional CNNs, will need to be trained specifically to learn texture features, a process that is not required if GCNNs are used. This is because GCNNs enforce

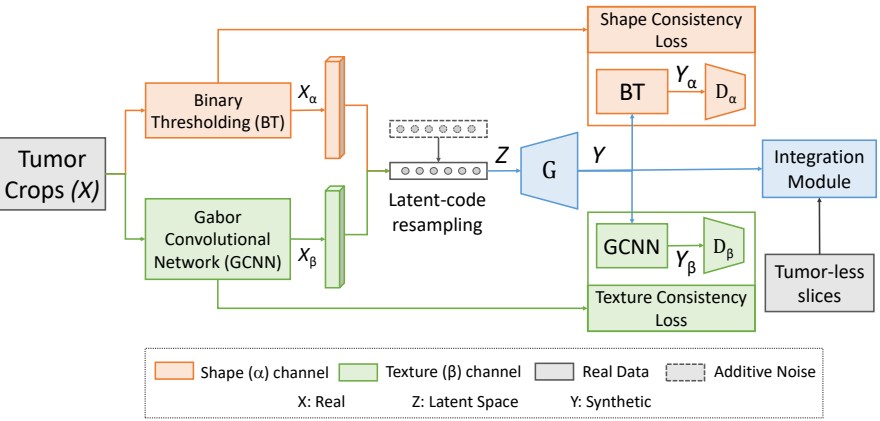

Figure 4: Proposed FeaD-GAN framework

Gabor-like properties to the layers of a CNN, thereby ensuring a texture representation. Additionally, GCNNs induce steerable properties into the CNN filters, making them invariant to scale and orientations (Luan et al., 2018). To design the GCNN, we implement Gabor Orientation Filters (GOFs) and perform convolution with filters of a standard CNN. Since the GOF equations are invariant, the only trainable parameters are the ones in the CNN which makes backpropagation possible in the GCNN. The standard set of Gabor filters is defined as:

$$g(u, v | f, \theta, \psi, \sigma) = exp\Big( - \frac{(u^2/\sigma_u^2 + v^2/\sigma_v^2)}{2} \Big) \times cos(2\pi f u + \psi) \qquad (1)$$

where, $u = xcos\theta + ysin\theta$ and $v = -xsin\theta + ycos\theta$, $x$ and $y$ are spatial positions, $\sigma_u$ and $\sigma_v$ are the spread of the filters, $f$ is the frequency of the sinusoid, $\theta$ is the orientation of the filters and $\psi$ is the phase-shift.

These features are then cast into latent space and re-sampled from a new distribution. Since we now have two distributions to sample from (instead of one in spatial domain), this protocol effectively increases the *apparent size of the dataset* by a power of two. Before feeding this sampled feature vector to Generator (G), we add random noise to this vector. This prevents the input to become entirely deterministic (like in Variational Autoencoders (VAE)) - thereby avoiding chances of mode collapse. Additionally, the noise also adds to the variety of input fed to G. Thus, we feed a semi-deterministic input vector ($z$) to the Generator, essentially picking on the benefits of both GANs and VAEs.

To ensure that the network learns both shape and textures, we design a loss function that is a combination of a shape consistency loss $\mathcal{L}_\alpha$ and a texture consistency loss $\mathcal{L}_\beta$. We train a separate discriminator between $X_\alpha$ and $Y_\alpha$ ($D_\alpha$) to obtain loss $\mathcal{L}_\alpha$. To calculate the $\mathcal{L}_\beta$ loss, we input the generated image to the GCNN ($Y_\beta$) and train a discriminator between real and fake textures ($D_\beta$). This aligns the probability distributions between real and fake textures. Furthermore, unlike traditional GANs, FeaD-GAN has both an added GCNN as well as a latent sampling protocol for noise input. Thus, we observe potentially decreased stability of FeaD-GAN as opposed to traditional GAN. To improve the stability,

we train both discriminators on Wasserstein distance. Wasserstein GANs (WGANs) have been shown to add stability to the GAN training (Arjovsky et al., 2017). They enforce a Lipschitz constraint which can sometimes result in overshoot, thus, we also apply gradient clipping and penalty function to the network (Gulrajani et al., 2017) to improve training stability. The shape consistency loss $\mathcal{L}_\alpha$ and texture consistency loss $\mathcal{L}_\beta$ is given in Eq 2 and 3 respectively.

$$\mathcal{L}_\alpha = \mathbb{E}_{y_\alpha \sim p_{Y_\alpha}}\left[\mathbf{D}_\alpha(y_\alpha)\right] - \mathbb{E}_{x_\alpha \sim p_{X_\alpha}}\left[\mathbf{D}_\alpha(x_\alpha)\right] + \mathbb{E}_{\hat{x}_\alpha \sim p_{\hat{X}_\alpha}}\left[||\nabla_{\hat{x_\alpha}}\mathbf{D}_\alpha(\hat{x}_\alpha)||_2\right] \qquad (2)$$

$$\mathcal{L}_\beta = \mathbb{E}_{y_\beta \sim p_{Y_\beta}}\left[\mathbf{D}_\beta(y_\beta)\right] - \mathbb{E}_{x_\beta \sim p_{X_\beta}}\left[\mathbf{D}_\beta(x_\beta)\right] + \mathbb{E}_{\hat{x}_\alpha \sim p_{\hat{X}_\beta}}\left[||\nabla_{\hat{x_\beta}}\mathbf{D}_\beta(\hat{x}_\beta)||_2\right] \qquad (3)$$

where, $Y$ is $G(Z)$, $Y_\alpha$ is $BT(Y)$ and $Y_\beta$ is $GCNN(Y)$. $Z$ is the feature vector sampled from $X_\alpha$ and $X_\beta$ cast onto the latent space, $\hat{X}$ terms are the weighted average between real and synthetic images ($\alpha$ for shape and $\beta$ for texture), $BT(.)$ is the Binary Thresholding operation, $GCNN(.)$ is the Gabor Convolutional Network output and $G$ is the Generator.

Finally, the overall loss $\mathcal{L}$ objective function to be minimized with $\mathcal{L}_\alpha$, $\mathcal{L}_\beta$ and gradient penalty is shown given in Eq 4.

$$\mathcal{L} = \lambda_\alpha \times \mathcal{L}_\alpha + \lambda_\beta \times \mathcal{L}_\beta \qquad (4)$$

where, $\lambda_\alpha$ and $\lambda_\beta$ are weights for the shape and texture consistency loss, respectively.

The generated tumors represent a synthetic version of R3. To recreate R1 and R2, we implement an integration block which fuses the generated tumors with pseudo-healthy brain slices (PHBs). These PHBs are the image slices that do not contain tumors. We used PHBs instead of publicly available healthy brain MRIs to have control over factors such as machine settings, normalization protocol etc. Control over these variables benefits us in creating better-quality synthetic R1 and R2s. For synthesizing the location and size of tumors, we generated a distribution of these features from the training data. From this we get a range of values in which the feature should lie. We randomly sampled values in this range such as the relative location, relative size, etc. of the generated tumor. This decides where to place the tumor and then applies post-processing filters to make the integration of the tumor into the brain smoother by using local edge-tapering 2D Gaussian filters.

## 4. Experimental Results

### 4.1. Description of dataset

For our analysis, we use two datasets, a private dataset collected at Emory University (Dataset A) and the publicly available The Cancer Imaging Archive (TCIA) (Clark et al., 2013) Glioblastoma Dataset (Dataset B).

**Dataset A.** For the private data, Dataset A, we assemble FLAIR MR images of a cohort of 25 patients with known 19/20 co-gain status. The cohort is divided into 14 control and 11 mutated patients. The control cohort has no 19/20 co-gain and the mutated has co-gain. This dataset contains about 9 images per patient that contain the tumor. Thus, both classes have close to 100-130 images each. Following the train-test split, this number becomes smaller, making it a very limited dataset.

**Dataset B.** For the public Dataset B, FLAIR MR images of a cohort of 165 patients are used. The cohort is divided into 134 control images and 31 mutated patients. The corresponding patient molecular data was extracted from The Cancer Genome Archive (TCGA) website (Tomczak et al., 2015). The genomic data was analyzed to extract 19/20 co-gain status. It was done by taking gistic data from TCGA and thresholding for each arm of the chromosome, if more than 25% of the genes are duplicated on both the long and short arms of chromosomes 19 and 20.

While Dataset B is not as scarce as Dataset A, it is heavily imbalanced. Thus, generated images using Dataset A signify the effectiveness of FeaD-GAN in learning discriminative features of very limited data, whereas, for Dataset B, the evaluation is primarily to combat data imbalance. The corresponding results are in Section 5.

### 4.2. List of Experiments

We perform an extensive set of experiments to evaluate every aspect of our proposed approach. The analysis is broadly divided into 4 parts:

*(a) To evaluate presence of mutation biomarkers in MR images:* We start by performing a baseline binary classification on whole slide images of Dataset A and B between control and mutated classes. We performed 10-fold cross validation on each of the datasets using multiple state-of-the-art classifiers, out of which ResNet18 was chosen because of its superior performance. The performance of other classifiers is shown in Appendix C2 and D1.

*(b) To characterize macroscopic features:* As described in Section 3.1 (refer Figure 2), after validating the presence of imaging features, we also perform experiments to quantify the impact of various macroscopic imaging features. The four representations of the data (R1-R4) in Figure 3 are used for classification. The results are reported in Table 1. Each of these results is the mean and standard deviation over 10-fold cross validation. Since Dataset B has heavy imbalance, we compare results using both Standard Data Augmentation (SDA: oversampling) and Custom Data Augmentation (CDA: FeaD-GAN).

*(c) To evaluate reproducibility of biomarkers:* To evaluate reproducibility, we train the classifier solely on synthetic images generated using FeaD-GAN and test this model on real images. The rationale behind this is that if the synthetic images accurately capture the discriminative features of imaging biomarkers, they should be able to distinguish between presence and absence of mutation. This doubles as a quantitative evaluation of how well FeaD-GAN is able to capture desired data distribution. The results for this evaluation are shown in Table 2. For qualitative evaluation, images generated by FeaD-GAN are shown in Figure 5. While this evaluation is primarily done on Dataset A because of its lack of samples, we also report a smaller set of evaluation on Dataset B in Table 2.

### 4.3. Results

In this evaluation, we used 80-20 data split for train-test and the data is divided patient-wise. For results in Table 1, we also use the same settings. The results are shown on two levels: patient-level (PL) and image-level (IL) classification. IL classifications are the results by considering each slice independently. PL classifications are obtained by computing the weighted mean of all images of a patient. Classifications are done using ResNet18. The representations R1-R4 are as described in Figure 3. Figure 5 shows the tumors generated

Table 1: Performance of ResNet18 (trained from scratch) over representations R1-R4 for both datasets. The values are Mean (Std) over 10-folds. SDA is Standard Data Augmentation (Oversampling) and CDA is Custom Data Augmentation (using FeaD-GAN). Here, RX: Representation from Figure 3, PL: Patient Level results, IL: Image Level results, ACC: Accuracy, SEN: Sensitivity, SPEC: Specificity, DIC: Dice Score

| Dataset | RX | ACC (PL) | ACC (IL) | SEN (IL) | SPEC (IL) | DIC (IL) |
|---------|----|----------|----------|----------|-----------|----------|
| *Dataset A* | R1 | 0.92 (0.08) | 0.89 (0.06) | 0.85 (0.07) | 0.95 (0.05) | 0.87 (0.08) |
| | R2 | 0.95 (0.03) | 0.92 (0.05) | 0.88 (0.06) | 0.98 (0.02) | 0.88 (0.08) |
| | R3 | 0.85 (0.08) | 0.80 (0.08) | 0.67 (0.09) | 0.84 (0.06) | 0.69 (0.09) |
| | R4 | 0.70 (0.06) | 0.68 (0.07) | 0.65 (0.09) | 0.75 (0.05) | 0.66 (0.08) |
| *Dataset B* *(SDA)* | R1 | 0.81 (0.09) | 0.78 (0.12) | 0.73 (0.08) | 0.86 (0.08) | 0.70 (0.10) |
| | R2 | 0.84 (0.08) | 0.78 (0.10) | 0.72 (0.06) | 0.88 (0.07) | 0.71 (0.10) |
| | R3 | 0.74 (0.09) | 0.73 (0.12) | 0.66 (0.10) | 0.77 (0.06) | 0.64 (0.13) |
| | R4 | 0.64 (0.10) | 0.62 (0.08) | 0.58 (0.06) | 0.70 (0.05) | 0.60 (0.08) |
| *Dataset B* *(CDA)* | R1 | 0.85 (0.05) | 0.84 (0.06) | 0.81 (0.06) | 0.90 (0.03) | 0.82 (0.08) |
| | R2 | 0.89 (0.05) | 0.86 (0.07) | 0.83 (0.06) | 0.96 (0.02) | 0.81 (0.08) |
| | R3 | 0.80 (0.08) | 0.78 (0.06) | 0.75 (0.12) | 0.86 (0.07) | 0.72 (0.09) |
| | R4 | 0.70 (0.04) | 0.66 (0.08) | 0.63 (0.10) | 0.74 (0.06) | 0.58 (0.13) |

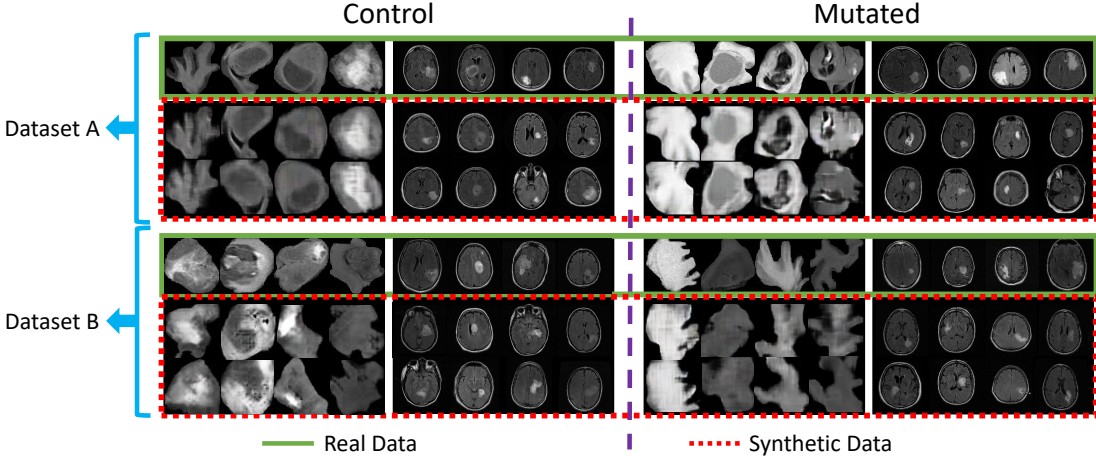

Figure 5: Tumor crops and Whole Slide Images generated using FeaD-GAN for both classes of Datasets A and B

using FeaD-GAN vs. real data. Figure 5 shows the WSIs generated using FeaD-GAN vs. real data. In Table 2, we report results on training a classifier using synthetic images and testing on real images for the four representations. Dataset A is fairly balanced whereas

Table 2: Mutation detection using only synthetic data generated using FeaD-GAN. Data is generated using the training set of Table 1 and tested on the corresponding real test set for all 10-folds. Here, RX: Representation from Figure 3, PL: Patient Level results, IL: Image Level results, ACC: Accuracy, SEN: Sensitivity, SPEC: Specificity, DIC: Dice Score

| Dataset | RX | ACC (PL) | ACC (IL) | SEN (IL) | SPEC (IL) | DIC (IL) |
|---|---|---|---|---|---|---|
| *Synthetic* *(Dataset A)* | R1 | 0.92 (0.09) | 0.88 (0.07) | 0.85 (0.08) | 0.94 (0.04) | 0.86 (0.08) |
| | R2 | 0.95 (0.03) | 0.90 (0.08) | 0.87 (0.07) | 0.98 (0.02) | 0.88 (0.09) |
| | R3 | 0.85 (0.08) | 0.82 (0.10) | 0.70 (0.08) | 0.86 (0.06) | 0.68 (0.08) |
| | R4 | 0.68 (0.06) | 0.66 (0.07) | 0.62 (0.07) | 0.74 (0.5) | 0.62 (0.08) |
| *Synthetic* *(Dataset B)* | R1 | 0.84 (0.05) | 0.82 (0.07) | 0.78 (0.08) | 0.89 (0.04) | 0.78 (0.10) |
| | R3 | 0.82 (0.06) | 0.79 (0.08) | 0.75 (0.09) | 0.90 (0.04) | 0.74 (0.06) |

for Dataset B, we used standard data augmentation (SDA) techniques to train. Using results from FeaD-GAN as a custom data augmentation (CDA) tool to balance the highly imbalanced Dataset B, we notice the performance improvement in both training on tumor crops and WSIs. The results are in Table 1 (Dataset B - SDA/CDA).

From Table 1, we note that representation R2 consistently gives best performance. We see that adding texture information to shapes (R3 from R4) gives a performance improvement of at least 10%, whereas providing location information further improves detection by approx. 10%. When adding global information, we find that there is no signifcant variation in performance, thereby it is currently not conclusive whether global features help or hinder detection. In Table 2 we see that the overall performance of R1-R4 is very similar to that in Table 1 for corresponding datasets - thereby suggesting that FeaD-GAN is able to accurately learn the features. It is possible that while learning texture, the GCNN learns some inherently correlated shape features as well. We still chose to add separate feature channels for added control in increasing the possible combinations of shape and texture to generate variety of images. To quantify that separate channels help learn more features than just texture channel, we ran experiments using only texture or only shape channel and used the generated images for classification. The texture channel results were compared to representation R3 and shape channel results were compared to R4 in Table 2. Dataset A was used for both comparisons. Accuracy by using only texture channel for synthetic images was 0.74 as opposed to 0.82 of Synthetic R3, whereas shape channel classification accuracy vs synthetic R4 is 0.59 vs 0.66.

## 5. Conclusions

From classification results of R1 from Table 1, we can conclude that imaging biomarkers of 19/20 co-gain are present. Furthermore, we notice that the configuration R2 gives best performance. However, it should also be noted the with such a small dataset, a difference in performance of 5% is not very significant. The impact can be caused either by global features, randomness in training or limitations of the classifier. Thus, we cannot conclude with confidence without further analysis whether or not global features impact detection of

co-gain. Noticing improvement from R4 to R3 and R3 to R2, we can conclude that texture and location are crucial. We note that data augmentation using FeaD-GAN gives better performance than oversampling. From Figure 5, we see that FeaD-GAN does a good job of generating quality tumor images. As to whether it captures the discriminating features, the evidence can be seen in Table 2, where synthetic images perform just as well as real images in mutation detection task. Based on our evaluations, we conclude that 19/20 co-gain imaging features exist, they are fairly consistent and can be reproduced. We also conclude that FeaD-GAN is an effective method to accurately generate subtle discriminatory features using very limited datasets, whilst serving as an effective data augmentation tool.

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

## Appendix A. Architectures of individual models:

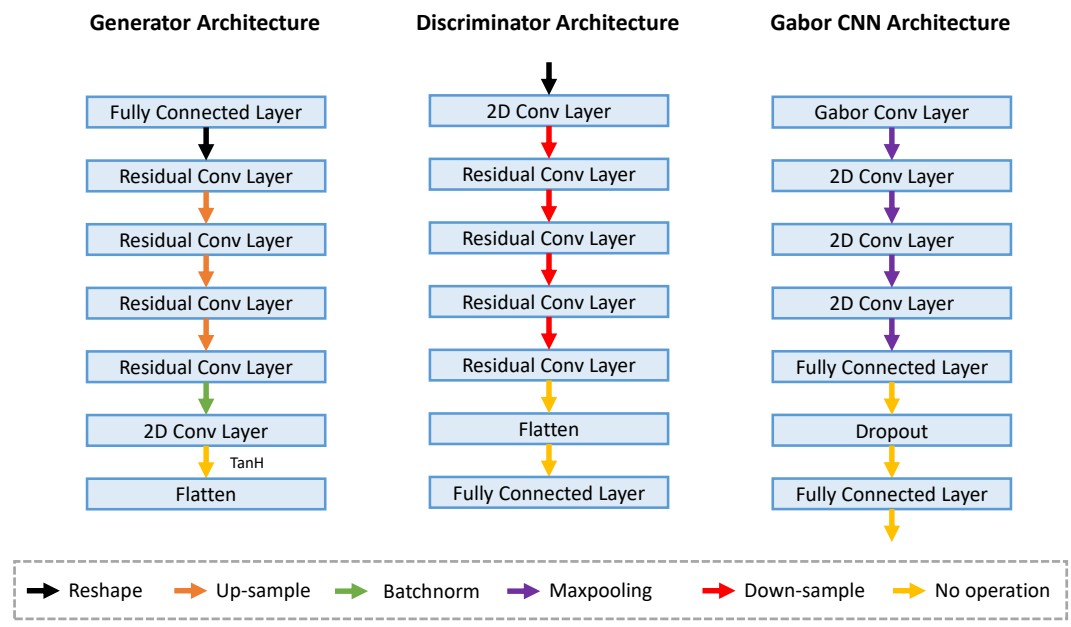

Figure 6: Architectures of Generator, Discriminator and Gabor CNN

## Appendix B. Skull stripping algorithm:

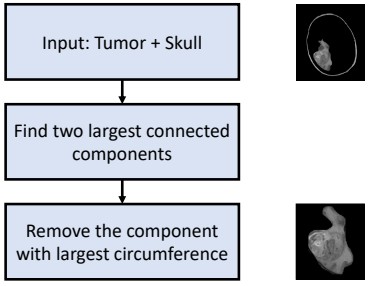

Figure 7: Pipeline for skull stripping

## Appendix C. Training Specifications

### C.1. For FeaD-GAN

For training FeaD-GAN, we initialize all the layers using a Kaiming initialization on account of its stable and superior convergence properties. The random noise added in the latent space is sampled from a standard normal distribution. All input data is normalized before training. Adam optimizer is used with a learning rate of 0.001 with a step decay protocol.

### C.2. For Classifiers

For InceptionV3 and AlexNet, we used pre-trained models and finetuned the last layer. However, in case of ResNet 18, we noticed an improvement in performance when trained from scratch. Optimizer used is Adam with a step LR scheduling protocol. The networks are trained using Binary Cross Entropy Loss.

Table 3: Results on multiple classifiers assessing visual presence of mutation. The data split is 80-20% and the table reports "mean (standard deviation)" over 10-folds

| Classifier | Dataset A | Dataset B |
|---|---|---|
| InceptionV3 | 0.82 (0.06) | 0.72 (0.07) |
| AlexNet | 0.74 (0.08) | 0.70 (0.09) |
| ResNet18 | 0.89 (0.04) | 0.78 (0.06) |

## Appendix D. Additional experiments:

### D.1. Choosing the single best imaging modality:

We conducted numerous experiments with FLAIR, T1-POST and DWI over many classifiers to check for consistently good performance. Our objective was to detect and recreate features from a single modality of data and found FLAIR to be the best single modality. The numbers are accuracy over 10 folds: mean accuracy (standard deviation). The results are shown on Dataset A, which is split into 80% training/validation and 20% testing.

Table 4: Results on multiple classifiers trained on various imaging modalities assessing visual presence of mutation. The data split is 80-20% and the table reports "mean (standard deviation)" over 10-folds

| Classifier | T1 Post | DWI | FLAIR |
|---|---|---|---|
| InceptionV3 | 0.58 (0.09) | 0.65 (0.07) | 0.82 (0.06) |
| AlexNet | 0.53 (0.10) | 0.58 (0.12) | 0.74 (0.08) |
| VGG19 | 0.55 (0.09) | 0.61 (0.11) | 0.77 (0.10) |
| ResNet18 | 0.61 (0.07) | 0.68 (0.06) | 0.89 (0.04) |

**D.2. Comparison between CNN and Standard Machine Learning classifiers:**

We ran experiments using Gray Level Co-occurence Matrix (GLCM) features with Support Vector Machine (SVM) with Radial Basis Function (RBF) Kernel and Random Forest (RF) classifier on whole images and only tumor images (R1 and R3 representations in Table 1) for Dataset A. These results were compared with classification using Convolutional Neural Networks (CNN). The results obtained are in the Table below:

Table 5: Classification performance of ResNet18 (CNN) as opposed to standard machine learning classifiers. The data split is 80-20% and the table reports "mean (standard deviation)" over 10-folds of Dataset A

| Input Images | Classifier | Accuracy | Sensitivity | Specificity | Dice Score |
|---|---|---|---|---|---|
| | SVM (RBF) | 0.72 (0.10) | 0.67 (0.08) | 0.75 (0.07) | 0.68 (0.08) |
| Whole Images | RF | 0.74 (0.11) | 0.68 (0.10) | 0.77 (0.06) | 0.70 (0.10) |
| | CNN | 0.89 (0.06) | 0.85 (0.07) | 0.95 (0.05) | 0.87 (0.08) |
| | SVM (RBF) | 0.67 (0.11) | 0.57 (0.09) | 0.68 (0.08) | 0.59 (0.10) |
| Tumor Crops | RF | 0.70 (0.12) | 0.58 (0.10) | 0.70 (0.06) | 0.59 (0.09) |
| | CNN | 0.80 (0.08) | 0.67 (0.09) | 0.84 (0.06) | 0.69 (0.09) |

