# OpenReview forum: "Feature Disentanglement to Aid Imaging Biomarker Characterization for Genetic Mutations"
_MIDL.io/2020/Conference — MIDL 2020_

### Official Review · AnonReviewer1 · 2020-03-10
**GAN-based model for learning from limited data using supervised feature disentanglement**

**Rating:** 2
**Confidence:** 5
**Recommendation:** Poster

**Summary:**

The paper proposes a GAN-based model that can be trained using limited data.  The core idea hinges on (1) a supervised disentanglement that explicitly capture the shape and image features separately and (2) a data-driven empirical distribution for the latent space that reduces the chances of mode collapse. Wasserstein distance is used for training stability. The proposed model is motivated by and evaluated within the context of detecting imaging biomarkers for genetic mutations.

**Strengths:**

- Embedding image and shape features into the latent space from which random samples are given to the generator is a reasonable idea to avoid mode collapse.
- Synthesized images (augmented data) are used to training classifiers, obtaining comparable performance with classifiers trained on real data.

**Weaknesses:**

- Data-driven embedding to avoid mode collapse has been proposed in GANs literature, e.g. BourGAN (https://arxiv.org/abs/1805.07674)
- Unjustified design choices. It is not clear why GCNNs are needed vs regular CNNs.
- The model relies on significant manual effort for tumor delineation to provide supervised disentanglement.
- Missing details, e.g. apparent size used for training, details for the integration module, how pseudohealthy data is generated, oversampling approach, training procedure.
- Missing ablation experiments.



**Justification Of Rating:**

The paper presents an interesting idea of data augmentation under limited data scenarios that is based on supervised disentanglement. Experimental results demonstrate that the trained model can reproduce imaging biomarkers relevant to the gene mutation. The paper is missing details and ablation experiments to study the impact of the shape/image disentanglement.

**Paper Type:**

both

**Questions To Address In The Rebuttal:**

- What is the apparent size used for training?
- Missing details about the integration module. How this module could synthesize variability in tumor location and orientation? Tumors tend to impinge the neighboring healthy tissue causing deformations, how is this handled/synthesized by the integration module? How is the integration module trained?
- Why GCNNs were used? what is GOFs?
- What is the oversampling approach used?
- What are the x_hat terms in (3) and (4)?
- How are the shape consistency loss errors backproped through the generator given the nondifferential binary thresholding block?
- The shape consistency loss bears some redundancy with the texture consistency loss since the boundary of the texture encodes some shape information (background has already been filtered out). What is the expected performance when only using one of these losses (ablation experiments)? This would empirically motivate the need for the proposed supervised disentanglement.
- How would this model handle the scenario of limited labeled data? coarse-level labels, e.g. bounding boxes rather than the accurate delineation of the tumor?


**Special Issue:**

no

---

> ### Author Response · Authors · 2020-03-28
> **Re: GAN-based model for learning from limited data using supervised feature disentanglement (Part 2)**
>
> Comment 8: What are the x_hat terms in (3) and (4)?
> Response: They are the gradient penalty terms [3] for training stability. The x_hat terms are the weighted average between real and fake data (shape in Eq (3) and texture in Eq (4)). They are passed through the discriminator and the expectation of the gradient norm is calculated. We noticed an error in the formula, where the last terms (gradient penalty) in Eq (3) and (4) should be as follows:
>
> E_(xhat~p_x ) [||∇_xhat {D_α (Xhat_α )}||_2 ] instead of E_(xhat~p_x ) [D_α (Xhat_α)]
>
> The above terms in Latex:
>
> \mathbb{E}_{\hat{x}_\alpha \sim {p}_{\hat{X}_\alpha}}\Big[{||\nabla_{\hat{x_\alpha}}}\mathbf{D_\alpha}(\hat{X}_\alpha)||_2 \Big]
> instead of
> \mathbb{E}_{\hat{x}_\alpha \sim {p}_{\hat{X}_\alpha}}\Big[{\mathbf{D_\alpha}(\hat{X}_\alpha)} \Big]
>
> We will include the description and edits in the final manuscript. These terms are added for improved stability in training [3].
> [3] Gulrajani, Ishaan, et al. "Improved training of Wasserstein GANs." Advances in Neural Information Processing Systems. 2017
>
> Comment 9: How are the shape consistency loss errors backproped through the generator given the nondifferential binary thresholding block?
> Response: We backprop only until the generator in the shape channel and not all the way through the binary thresholding block. We backprop all the way through the GCNN in the texture channel.
>
>
> Comment 10: The shape consistency loss bears some redundancy with the texture consistency loss since the boundary of the texture encodes some shape information (background has already been filtered out). What is the expected performance when only using one of these losses (ablation experiments)? This would empirically motivate the need for the proposed supervised disentanglement.
> Response: While it is true there is some redundancy between extracted features, it is to be noted that shape information obtained while extracting texture information are correlated to each other. We wanted added control in increasing the possible combinations of shape and texture to generate variety of images. This justifies the separate independent channels. We ran experiments using only texture or only shape channel and used the generated images for classification. The 10-fold accuracy was then compared with the Synthetic R3 (only tumor crops) and R4 (binary tumor crop mask) (refer Figure 3, Table 2-Dataset A). For reference, Synthetic R4 was obtained by binary thresholding of Synthetic R3. We will add a detailed analysis in the final manuscript.
> Synthetic R3 vs using only texture channel (Dataset A): 0.82 vs 0.74
> Synthetic R4 vs using only shape channel (Dataset A): 0.66 vs 0.59
>
>
> Comment 11: How would this model handle the scenario of limited labeled data? coarse-level labels, e.g. bounding boxes rather than the accurate delineation of the tumor?
> Response: We have not conducted experiments on coarse-level labels yet. Based on literature that used texture descriptors for tumor delineation [4, 5], we expect the texture analysis done by GCNN should tackle such issues. In addition, we can add a foreground extraction module if needed. However, this is yet to be analyzed. Our current dataset (A) is limited and the model is able to tackle that very well.
> [4] Alobaidli, S., et al. "The role of texture analysis in imaging as an outcome predictor and potential tool in radiotherapy treatment planning." The British journal of radiology 87.1042 (2014): 20140369.
> [5] Tong, Jijun, et al. "MRI brain tumor segmentation based on texture features and kernel sparse coding." Biomedical Signal Processing and Control 47 (2019): 387-392.

---

> > ### Comment · AnonReviewer1 · 2020-04-03
> > **Thanks for addressing my comments**
> >
> > In light of the authors clarifications and discussions, I am changing my rating to a weak accept.

---

> ### Author Response · Authors · 2020-03-28
> **Re: GAN-based model for learning from limited data using supervised feature disentanglement (Part 1)**
>
> The authors would like to thank the reviewer for the insightful comments. This will help us write a better manuscript. We would like to address the comments and questions in the review. We will make the necessary edits in the final manuscript.
>
> Comment 1: Manual tumor delineation:
> Response: (As explained in Reviewer 2’s Point 4): This was done because using an automated technique is not 100% accurate and any errors in tumor segmentation would trickle down to the rest of the representations (Figure 3) and thus affect the results of both classifier and FeaD-GAN.
>
> Comment 2: Ablation study:
> Response: We have images generated by FeaD-GAN by isolating the texture and shape channel. While these provide qualitative analyses, we can use these images for detection and compare their performance with the results in the paper (Table 1) for quantitative analysis (refer comment # 10). We already have results of various imaging modalities (T1-POST, DWI, FLAIR) and multiple (state-of-the-art) classifiers in mutation detection which we did not add to the submission. We have comparison of CNNs with handcrafted features for mutation detection. These results are shown in responses to Reviewers 1 and 2 and we plan to add them in the paper.
>
> Comment 3: Data-driven embedding to avoid mode collapse has been proposed in GANs literature, e.g. BourGAN
> Response: Yes, data-driven embedding has been proposed in literature. In addition to avoiding mode collapse, our goal here is to generate diverse synthetic images from very limited (as small as 80 samples per class in Dataset A) and imbalanced dataset (31 vs 134 patients in mutated and control classes, respectively, in Dataset B). The independent feature channels provide a larger search space and added control over generated data by influencing how each feature is sampled before embedding.
>
> Comment 4: What is the apparent size used for training?
> Response: We have 80 training images per class for Dataset A. Typically features will be resampled from the latent representation of these 80 samples. In our case, we will have 80 shape samples and 80 texture samples (plus random noise), so we have >6400 data points to resample from. In terms of probability distribution, we now have two prior distributions (plus random noise) to sample from instead of one as in case of standard sampling.
>
> Comment 5:
> (a) Missing details about the integration module. How this module could synthesize variability in tumor location and orientation? How is the integration module trained?
> Response: For synthesizing the location and size of tumors, we generated a distribution of these features from the training data. From this we get a range of values in which the feature should lie. We randomly sampled values in this range such as the relative location, relative size etc. of the generated tumor. This decides where to place the tumor and then applies post-processing filters. We will add these details to the paper.
>
> (b) Tumors tend to impinge the neighboring healthy tissue causing deformations, how is this handled/synthesized by the integration module?
> Response: Currently, we chose Pseudo-healthy images. These are essentially the tumor-less slices from the patients in the training set that had some residual effect of tumor growth. Thus, we tried to get as close as possible to the deformation using the data we had.
>
> Comment 6: Why GCNNs were used? what is GOFs?
> Response: We wanted to ensure only texture data was embedded in the feature vector. GCNNs [2] ensure the texture representation of the data. Traditional CNNs will need a task and be trained to specifically learn texture features. GCNNs enforce Gabor-like properties to the layers of a CNN, thereby ensuring a texture representation. Additionally, GCNNs induce steerable properties into the CNN filters, making them invariant to scale and orientations [2].
> GOFs are Gabor Orientation Filters. We will add these details in the main paper.
> [2] Luan, Shangzhen, et al. "Gabor convolutional networks." IEEE Transactions on Image Processing 27.9 (2018): 4357-4366.
>
> Comment 7: What is the oversampling approach used?
> Response: We used class weights and random duplication from the minority class.

---

### Official Review · AnonReviewer3 · 2020-03-14
**A GAN approach to improve genetic mutation prediction.**

**Rating:** 2
**Confidence:** 4
**Recommendation:** Poster

**Summary:**

This paper utilizes brain tumor MRI images to predict molecular biomarkers utilizing deep learning. The authors attempt to address two major issues common in such tasks: Imbalanced dataset and lack of training data.  Predicting such biomarkers utilizing MRI imaging can improve patient management.  Both deep learning and and texture features are utilized.

**Strengths:**

A novel approach is used toward data augmentation. The paper utilizes a publicly available dataset in addition to a private dataset.  Multiple data representation approach is attempted comparing classical texture features  and deep learning classification schemes.

**Weaknesses:**

Accuracy is not a good metric to use when dealing with imbalanced datasets. The authors should consider reporting weighted F1, precision and recall.
Typo in Introduction, Line 8: “precise approach. the concurrent”.
It will be ideal to compare the method proposed with classical augmentation approaches, like rotation, scaling, and translation.
How the tumor and skull image was obtained.


**Justification Of Rating:**

The metrics utilized to prove the performance of the system are not adequate. The authors need to provide additional information with respect to the methodologies utilized in all the steps of this paper.

**Paper Type:**

methodological development

**Questions To Address In The Rebuttal:**

Please see weaknesses. The authors need to utilize better metrics especially in datasets with high imbalance.

**Special Issue:**

no

---

> ### Author Response · Authors · 2020-03-28
> **Re: A GAN approach to improve genetic mutation prediction.**
>
> The authors would like to thank the reviewer for the insightful feedback. We would like to address some of the concerns pointed out in the review. We will make appropriate changes in the final manuscript.
>
> Comment 1: Accuracy is not a good metric to use when dealing with imbalanced datasets. The authors should consider reporting weighted F1, precision and recall.
> Response: Tables 1 and 2 in the Results (Section 5) have already included the commonly used evaluation measures in the medical community: Sensitivity (SENS), Specificity (SPEC) and Dice coefficient (DIC). They are all related with precision, recall and F1. We have conducted these experiments using 10-fold cross-validation for bolstering performance evaluation.
>
> Comment 2: Typo in Introduction, Line 8: “precise approach. the concurrent”.
> Response: We will correct this error
>
> Comment 3: It will be ideal to compare the method proposed with classical augmentation approaches, like rotation, scaling, and translation.
> Response: In Table 2 (Section 5), we have included two types of data augmentations: Standard Data Augmentation (SDA) and Custom Data Augmentation (CDA). SDA includes the Data Augmentation (DA) technique of resampling the dataset for balancing it. We refrained from using rotation, translation and scaling. This is because Table 1 implies importance of location in mutation prediction. Techniques such as rotation tamper with this information and may give us incorrect predictions.
> Additionally, we also conducted experiments using DA tools such as rotation, scaling etc. and did not see a significant improvement in performance. We included these results as well for comparison below. We believe the improvement in performance due to rotation, scaling etc. are better in R3 (than R1) because the location information has been removed. The values shown are for average Dice (F1) scores over 10 folds similar to Table 1: mean value (standard deviation)
> ----------------------------------------------------------------------------------------------
> Dataset B: (Full Images, R1)
> Data Augmentation using oversampling (Table 1): 0.70 (0.10)
> Data Augmentation using rotation, shear, scaling etc.: 0.71 (0.12)
> Data Augmentation using FeaD-GAN (Table 1): 0.82 (0.08)
>
> Dataset B: (Tumor Crops, R3)
> Data Augmentation using oversampling (Table 1): 0.64 (0.13)
> Data Augmentation using rotation, shear, scaling etc.: 0.66 (0.10)
> Data Augmentation using FeaD-GAN (Table 1): 0.72 (0.09)
> ----------------------------------------------------------------------------------------------
>
> Comment 4: How the tumor and skull image was obtained
> Response: The skull image was manually delineated (R2 in Figure 3). This was done manually because using an automated technique is not 100% accurate. Brain tumor segmentation is still an (actively) developing field. Any errors in automated segmentation would trickle down to the rest of the representations (Figure 3) and thus affect the results of both: classifier and FeaD-GAN. The tumor image (R3 in Figure 3) was obtained using skull stripping technique as shown in Figure 7 in Appendix.

---

### Official Review · AnonReviewer5 · 2020-03-19
**The paper aims to predict the  to predict 19/20 co-gain status using feature Disentanglement with  Generative Adversarial Networks**

**Rating:** 3
**Confidence:** 5
**Recommendation:** Poster

**Summary:**

The authors use the feature Disentanglement (FeaD-GAN) technique for generating synthetic images and re-sample from a pseudo-larger data distribution to generate synthetic images from limited data.  This is a good idea to overcome the lack and unbalance data.
Three experiments were considered in this study are:
1) To evaluate the presence of mutation biomarkers in MR images,
2) To characterize macroscopic features,
3)To evaluate the reproducibility of biomarkers.
The classifier accuracy of 19/20 co-gain status is good but not validated to be significant.

**Strengths:**

Authors used conventional features like the texture and shape features with FeaD-GAN framework.
The results are achieved the highest result when they considered the texture, shape, and location of the tumor.

**Weaknesses:**

The authors applied the classifier model just on FLAIR while the TCIA datasets consist of four MRI sequences.
Many minor typos should be avoided. and all the symbols should be defined.
The baseline should be not only CNN but also the texture (LoG, GLCM,..etc.) and shape features  as in the radiomic or radiogenomics analysis.

**Detailed Comments:**

in previous sections

**Justification Of Rating:**

Application is needed in the medical field. In addition, this paper addresses the real challenges like lack and unbalance data and how the proposed work can avoid these issues using the advantage of deep learning with texture and shape features.

**Paper Type:**

both

**Questions To Address In The Rebuttal:**

1- We suggest the authors consider the baseline as following, use the texture, shape, and location of the tumor as input to the random forest or SVM and predict the 19/20 co-gain status, then compare based on synthetic data..etc.
2-What is the benefit that GAN applied on pre identified ROI on 2D, many works in this context should be cited like https://rd.springer.com/chapter/10.1007/978-3-030-40124-5_4


**Special Issue:**

yes

---

> ### Author Response · Authors · 2020-03-28
> **Re: The paper aims to predict the to predict 19/20 co-gain status using feature Disentanglement with Generative Adversarial Networks**
>
> The authors would like to thank the reviewer for the insightful feedback. We would like to address the comments and concerns of the reviewer. We will make appropriate changes in the final manuscript.
>
> Comment 1: The authors applied the classifier model just on FLAIR while the TCIA datasets consist of four MRI sequences.
> Response: We have conducted numerous experiments with FLAIR, T1-POST and DWI over many classifiers to check for consistently good performance. Our objective was to detect and recreate features from a single modality of data and found FLAIR to be the best single modality. We will include these results from each modality in the paper. The numbers are accuracy over 10 folds: mean accuracy (standard deviation). The results are shown on Dataset A, which is split into 80% training/validation and 20% testing.
>
> Dataset A	    T1 Post	         DWI	         FLAIR
>
> InceptionV3	   0.58 (0.09)	    0.65 (0.07)	    0.82 (0.06)
> AlexNet	           0.53 (0.10)	    0.58 (0.12)	    0.74 (0.08)
> VGG19	           0.55 (0.09)	    0.61 (0.11)	    0.77 (0.10)
> ResNet18	   0.61 (0.07)	    0.68 (0.06)	    0.89 (0.04)
>
> Comment 2: Many minor typos should be avoided and all the symbols should be defined
> Response: We will make the necessary edits
>
> Comment 3: The baseline should be not only CNN but also the texture (LoG, GLCM,..etc.) and shape features as in the radiomic or radiogenomics analysis
> Response: We ran experiments using GLCM features with SVM (RBF Kernel) and random forest (RF) classifier on only tumor images (R3 in Table 1 for Dataset A). We achieved an average detection performance of 0.67 and 0.70 using SVM and RF, respectively. Compared to this, we achieved an average detection performance of 0.80 using CNNs. For whole images (R1 in Table 1 for Dataset A), we achieved a performance of 0.72 (SVM) and 0.74 (RF) as opposed to 0.89 with CNNs. We will add detailed results with all metrics (as in Table 1 and 2) as a comparison with CNNs in the final manuscript.
>
> Comment 4: What is the benefit that GAN applied on pre identified ROI on 2D, many works in this context should be cited like [1]
> Response: The paper suggested by the reviewer bolsters our motivation for the current approach. We will add this to our paper. The benefit of pre-identified ROI on 2D has some important benefits. In addition to [1], automated annotation does not have perfect performance and any errors will trickle down to the rest of the model (both GAN and the analysis in Figure 3).
> [1] Chaddad, Ahmad, et al. "Deep radiomic features from MRI scans predict survival outcome of recurrent glioblastoma." arXiv preprint arXiv:1911.06687 (2019).

---

### Meta-Review · Area_Chair1 · 2020-04-06
**MetaReview of Paper250 by AreaChair1**

**Rating:** 3
**Recommendation For Accepted Papers:** Poster

**Metareview:**

The papers presents an original application of feature disentanglement GANs to characterize the manifestation of 19/20 co-gain in MRI of glioblastoma patients. The proposed approach is used as data augmentation to overcome the problem of limited and unbalanced data related to this genetic mutation. Experiments show that the model can reproduce imaging biomarkers relevant to 19/20 co-gain. In their rebuttal, authors have answered the reviewers' main concerns.


**Paper Type:**

both

**Special Issue:**

no

---

### Decision · Program_Chairs · 2020-04-11

Accept